:③: PLOS | ONE

# Perceptions of risk and influences of choice in pregnant women with obesity. An evidence synthesis of qualitative research

Sophie Relph[1]*, Melissa Ong[2], Matias C. Vieira[1], Dharmintra Pasupathy[1], Jane Sandall[1]

1 Department of Women and Children's Health, School of Life Course Sciences, Faculty of Life Sciences and Medicine, King's College London, London, United Kingdom, 2 Guy's, King's and St Thomas' School of Medical Education, King's College London, London, United Kingdom

* sophie.relph@kcl.ac.uk

**Data Availability Statement:** All relevant data are within the manuscript and its Supporting Information files.

## Abstract

### Background

Between 7–35% of the maternity population are obese in high income countries and 1–40% in lower or middle-income countries. Women with obesity are traditionally limited by the choices available to them during pregnancy and birth because of the higher risk of complications. This evidence synthesis set out to summarise how women with obesity's perceptions of pregnancy and birth risk influence the care choices that they make.

### Methods

A search of medical and health databases for qualitative studies written in the English language, published Jan 1993—April 2019 and reporting on pregnant women with obesity's perception of risk and influence of pregnancy and birth choices. Data was extracted by two reviewers onto a questions framework and then analysed using a thematic synthesis technique. Confidence in the qualitative findings was assessed using GRADE-CERQual.

### Results

23 full texts were included. The common themes on perception of risk were: 'Self-blame arising from others' stereotyped beliefs ', 'Normalisation', 'Lack of preparation', 'Fearful acceptance and inevitability' and 'Baby prioritised over mother'. For influence of choices, the themes were: 'External influences from personal stresses', 'Restrictive guidelines', 'Relationship with healthcare professional' and 'Perception of Risk'.

### Conclusions

Evidence on what influences women with obesity's pregnancy choices is limited. Further research is needed on the best methods to discuss the risks of pregnancy and birth for women with obesity in a sensitive and acceptable manner and to identify the key influences when women with obesity make choices antenatally and for birth planning.

**Funding:** MCV is supported by CAPES (BEX 9571/13–2; https://www.capes.gov.br/). SR is supported through a Guy's and St Thomas' charity grant https://www.gsttcharity.org.uk/). JS and SR are supported by the National Institute for Health Research (NIHR) Applied Research Collaboration South London at King's College Hospital NHS Foundation Trust. The views expressed are those of the authors and not necessarily those of, the NIHR or the Department of Health and Social Care. DP is funded by Tommy's Charity (https://www.tommys.org/) and NIHR Biomedical Research Centre at Guy's & St Thomas' NHS Foundation Trust and King's College London (http://www.guysandstthomasbrc.nihr.ac.uk/). The funders had no role in study design, data collection and analysis, decision to publish, or preparation of the manuscript.

**Competing interests:** The authors have declared that no competing interests exist.

## Introduction

Prevalence estimates of maternal obesity (pre-pregnancy body mass index (BMI) $>30kg/m^2$) vary across high income countries with an estimated 7.1% of the pregnant population having obesity in Poland (the lowest European rate), 21% in the UK and 31.8% in the USA[1, 2]. In lower and middle-income countries, rates of obesity in women of reproductive age vary between 1% in Ethiopia to 39.6% in Egypt[2]. Obesity is associated with adverse outcomes in pregnancy, but a secondary analysis of a large UK study (Birthplace, 2011) found that over 60% of women with obesity but no other comorbidities or complications prior to term labour, had vaginal births without maternal complication or intervention and over 95% had births without neonatal unit admission or perinatal death[3].

The recently published UK Royal College of Obstetricians and Gynaecologists (RCOG) guidance on Care of Women with Obesity in Pregnancy recommends that pregnant women with obesity be integrated into all antenatal clinics, that otherwise low-risk, multiparous, women with obesity can be offered choice of care setting for birth in obstetric or midwifery-led units and that all women should have informed discussions which consider their wishes when planning for labour and birth[4, 5]. Older UK guidance previously advised that women with BMI$>35kg/m^2$ should give birth in consultant-led birthing centres and that women with BMI 30-35kg/m$^2$ should be individually assessed when deciding place of birth[6]. Internationally, neither the International Federation of Gynaecology and Obstetrics, the American College of Obstetricians and Gynecologists nor the World Health Organisation make recommendations on place of care, lead health professional or maternal choice during pregnancy for women with obesity.

A UK study (2016) examining influences of birthplace choices in healthy women with straightforward pregnancies found that women were influenced by the information that they received, including about the right to choose. Women's choices were also influenced by previous birth experiences, views of family, friends or healthcare professionals (HCPs) and the women's beliefs about risk and safety[7]. This was not specific to women with obesity in pregnancy, who are often restricted by the choices of care they are offered.

## Methods

A qualitative synthesis was conducted which aimed to summarise the current qualitative evidence on how women with obesity's perceptions of pregnancy and birth risk influence their care choices.

The protocol for this systematic review and thematic synthesis was pre-planned and registered on PROSPERO (reference CRD42018091990)[8]. This study has been reported in line with the recommendations of the ENTREQ statement (Enhancing Transparency in REporting the synthesis of Qualitative research)[9]. The ENTREQ checklist can be viewed in S1 Table.

### Search Strategy

A comprehensive search was pre-planned using the SPIDER tool (S2 Table)[10] and conducted using the BNI, CINAHL, EMBASE, Medline, Web of Science and Cochrane databases. See S3 Table for the full search terms.

The search was limited to publication year January 1993- April 2019 and to English language studies. 1993 was chosen because this was the date the UK Changing Childbirth report was published. This recommended that future policy for maternity services be centred around the wants and needs of women[11]. Only primary qualitative research studies were included. Conference abstracts were excluded because of insufficient detail for analysis. Reference lists of review articles were screened for additional relevant papers.

## Inclusion and exclusion criteria

Studies were included if they reported data on perceptions of antenatal and/or labour risks or influences of choices during pregnancy or birth, amongst women with obesity. Studies were excluded if they reported only on the views of healthcare professionals or family members, collected data more than six months postnatally (because of the risk of recall bias) or only described postpartum experiences (e.g. breastfeeding).

## Study selection

Following removal of duplicates, the titles and abstracts were screened against the inclusion and exclusion criteria by SR and MO. The remaining full texts were then screened against the same criteria, and for extractable primary and secondary construct data which answered the following questions:

1. How do women with obesity perceive obstetric risks and discussions about risks during pregnancy?

2. What influences the non-lifestyle choices that women with obesity make during pregnancy?

Only quotations or interpretations which were directly inferred from participants with obesity were synthesised, from studies which also included women in other weight ranges.

Study selection was conducted independently by SR and MO, who subsequently agreed the final list. Since the list of included studies was not large (n = 23), a comprehensive sampling strategy involving data extraction from all studies was deemed feasible.

## Analytical strategy

Following an initial familiarisation of the included literature and consultation of the RETREAT guidance for choosing an evidence synthesis methodology[12], we chose to undertake a two-step coding process. First, data was coded onto a question framework (S4 Table). This allowed the data to be sorted into the multiple question components of the review. Following this, a thematic synthesis technique was used to code inductively, describe and analyse the data within the question framework[13].

SR and MO independently generated inductive codes from the primary and secondary data constructs within the 'Findings' section of each paper, then met to agree the final codes. SR imported the data and codes onto the pre-agreed framework in NVivo (v12). SR and MO then reviewed the codes within each aspect of the framework, developing descriptive inductive themes. Finally, themes were explored within and across framework components, searching for similarities, conflicts and explanations, and amalgamating the data to produce synthesised findings.

## Quality assessment

The GRADE-CERQual approach was adopted to assess the level of confidence that the review findings were reasonable representations of the phenomenon of interest [14, 15]. With this approach, each finding is assessed as having no or very minor concerns, minor concerns, moderate concerns or serious concerns for: methodological limitations, the consistency across multiple contexts (coherence), relevance to the review question and adequacy/richness of evidence. For this review, methodological limitations of included primary studies were assessed independently by SR and MO, using the CASP Qualitative checklist[16, 17]. The overall CASP score was then agreed. Studies with serious methodological concerns were then excluded from the study. Following assessment of each component, overall conclusions

on the confidence in the evidence (very low, low, moderate or high confidence) are then drawn for each review finding. High confidence was applied to a review finding which had no or only one minor concern regarding the contributory components. Moderate confidence was determined where a review finding had one moderate or more than one minor concerns in the components. An overall rating of low confidence was determined by one component having serious concerns, or the at least two components having moderate concerns. A score of very low confidence was given to study findings which had more than one component with serious concerns.

# Findings

## Characteristics of included studies

The 2,226 studies identified from electronic searches and one study identified from hand-searching references were screened. After application of the inclusion and exclusion criteria, 23 studies were included in the final review (Fig 1, S5 Table).

The characteristics of the included studies have been summarised in S6 Table. Only seven studies set out to explore either of the questions of this review[18–24]. Most studies set out to document women's experiences of healthcare when obese and pregnant.

## Quality assessment

With regards to the CASP-guided methodological appraisal, we have serious concerns regarding the reported methodology of one study[25], moderate concerns regarding three studies [26–28] and minor or no concerns regarding the methodology of the remaining 19 studies [18–20, 23, 24, 29–41]. See S7 Table for further detail on the appraisal of each included study. The study with serious methodological concerns was then excluded from further analysis[25].

The results of the GRADE-CERQual appraisal of each synthesis finding are summarised in Table 1, reported sequentially as they present through the continuum of counselling. S8 Table details the fully explained GRADE-CERQual assessment, including all data which contributed to the findings.

## Study findings—Themes for perceptions of risk during pregnancy and birth

Five major themes were identified to answer the first study questions on how pregnant women with obesity perceive risk during pregnancy. The key findings are summarised in Table 1 and Fig 2. Table 2 lists the types of risks discussed in the included papers.

**Self-blame arising from stereotyped beliefs of others.** Women exposed to behaviours arising from stereotyped beliefs of their healthcare professionals, feel blamed and blame themselves for potential risks associated with their obesity. Data to support this finding was from women with all classes of obesity.

Women with obesity perceive that healthcare providers make assumptions about them based on pre-conceived ideas[21, 33]. They feel judged for their weight and are fearful of consultations because of the perceived stigma around being obese [31, 33]. This led to perceived over-inflation of the risk likelihood [18, 30, 31, 33]. In only one study, women believed it was the increased weight which inflated the risk and did not blame the provider's stigma [35]. As a result of these stereotyped attitudes and perceived judgements, women felt that they were 'boxed in' with other women affected by obesity [18, 33, 40], penalised for their weight [33, 34] and blamed for complications which arose [29–31, 33].

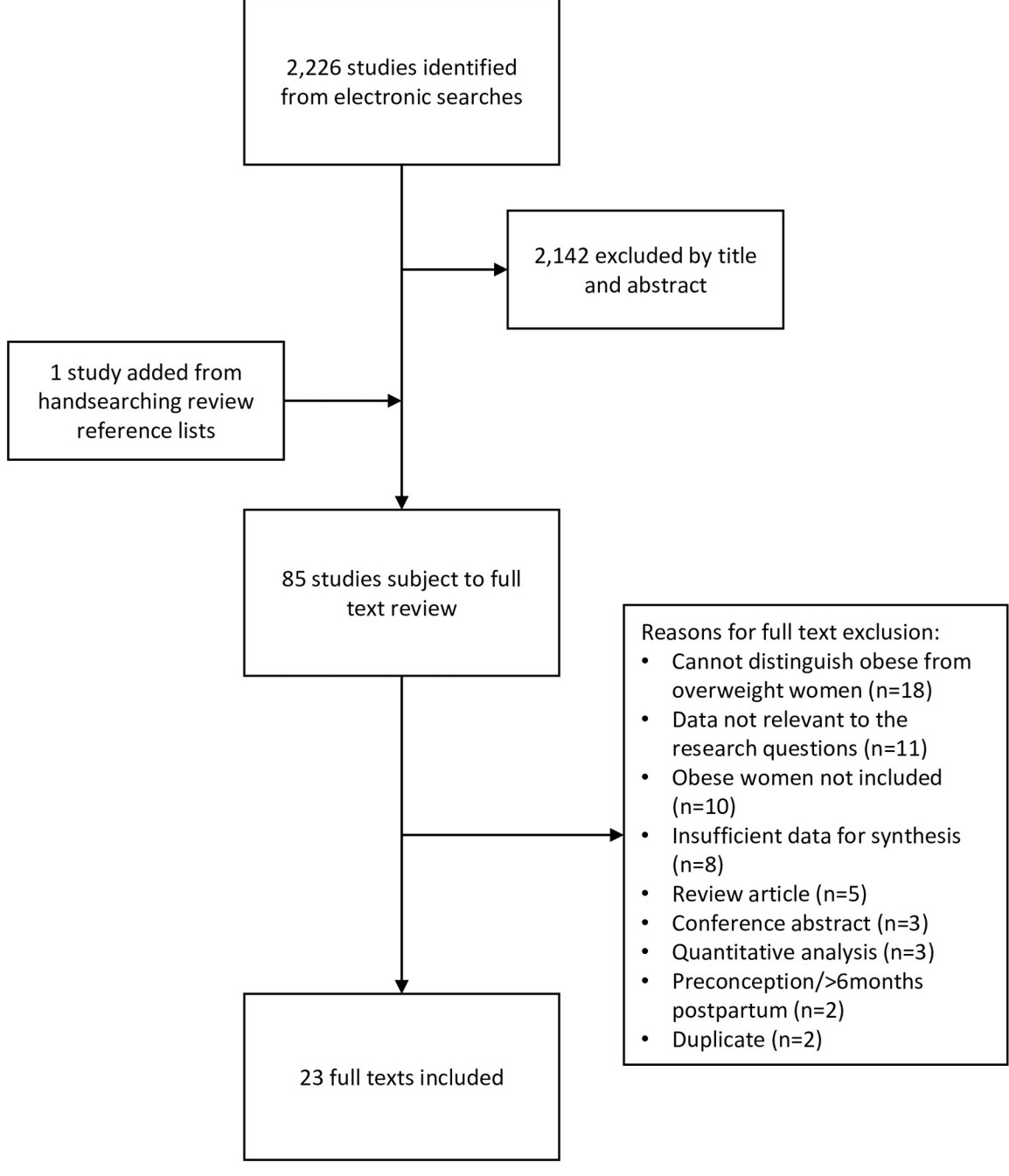

**Fig 1. Study selection process.**

Only one woman stated the opposite, because the counselling was "backed up by evidence" [41]. Another woman appreciated being identified within a group, because it made her feel that she was not at fault; risk happened to other women too [29].

**Normalisation of risks.** Whilst any woman with obesity might have altered perceptions of risk due to the external influence of stereotyped thought, the following three themes refer to distinctly different ways in which women approach risk in pregnancy, which appears to be heavily influenced by the women's experience of antenatal counselling [29, 32, 33, 37–39].

**Table 1. Summary of qualitative findings.**

| Qualitative Finding | CERQual assessment and explanation |
|---|---|
| Women felt that health-professionals pre-conceived stereotyped beliefs regarding their weight led to over-inflated presentations of risk | Low confidence: Minor methodological concerns, the interpretative link between healthcare professionals' stereotyped beliefs and over-inflation of risk is only adequately supported in two studies. The perceived over inflation of risk could be explained by the real risk of complications [18, 21, 30, 31, 33, 35, 41]. |
| Women felt penalised for their weight when health-professionals 'boxed them in' with other women affected by obesity. | High confidence: Minor methodological concerns, but the finding is coherent with adequate examples of rich data to support it [18, 33, 34, 40]. There is one example where this finding is refuted [41]. |
| An insensitive or stigmatising approach to counselling by health professionals led women to feel ashamed of their weight and blamed by the clinician or themselves for complications which arose. | High confidence: There are minor methodological concerns, but the finding is well supported from adequate data which is sufficiently rich [29–31, 33, 41]. |
| Most women and healthcare professionals avoid counselling regarding risks in pregnancy associated with obesity. | Moderate confidence: There are 2 papers with minor methodological concerns and are a few conflicting examples (women were counselled regarding risk and aware of potential complications) [19, 20, 29, 32, 36–38, 41]. |
| The way women with obesity perceived risk in pregnancy was heavily influenced by the nature or antenatal counselling received (or lack thereof). | Moderate confidence: There are minor methodological concerns and the data are not sufficiently rich in all examples to support this interpretative finding [29, 32, 37–39]. |
| A lack of counselling on potential complications of pregnancy causes women to feel unprepared or shocked when presented with risk. | High confidence: There are minor methodological concerns but coherent, adequate and relevant data which supports the finding [18, 19, 29, 37]. |
| A lack of discussion regarding the potential complications of pregnancy provides false reassurance to some women with obesity | Moderate confidence: Whilst there are enough cogent and relevant data from good quality studies, this is only rich to support this interpretative finding in two [20, 36, 38]. |
| Women who normalise potential risks in pregnancy do this in response to a belief that risks are either unrelated to their obesity, or to their perception that they themselves are healthy. | Moderate confidence: We have moderate methodological concerns in one of the four papers. There are adequate data which supports this finding, there are also disconfirming examples where women are well informed but deny the risk because of misinformation or denial [19, 21, 27, 33]. |
| Women who accept the potential for pregnancy risks proceed through pregnancy with anxiety or fear for the occurrence of complications. | Moderate confidence: Cogent and relevant data to support this finding, moderate methodological concerns and disconfirming cases where informed women manage risk pragmatically [19, 20, 27, 31–33, 40]. |
| Some women, who accept the potential for risk in pregnancy, consider such risks to be inevitable and their occurrence to be out of their control. | Low confidence: This finding is only supported by data from women with body mass index above 40kg/m$^2$ in 2 studies [21, 23]. |
| Some women with obesity often felt forgotten about during their antenatal care, with the needs of their unborn baby often prioritised above their own needs. | High confidence: This finding is well supported with adequate, relevant and cogent data with only minor concerns regarding the methodology of studies [18, 30, 38]. |
| Stresses with women's family and professional lives influence choices that they made regarding their antenatal care. | Low confidence: There are cogent data to support this finding with only minor concerns regarding methodology however, the finding is only supported by relevant data from two studies [18, 33]. |
| Relationships with healthcare providers which were perceived negatively by the women made them feel as if they had no choice in pregnancy and birth. | Moderate confidence: This finding is well supported by relevant, cogent data with only minor methodological concerns; however, the data only supports the explanatory portion of this finding in 2 studies [26, 30, 31, 33, 41]. |

*(Continued)*

**Table 1.** (Continued)

| Qualitative Finding | CERQual assessment and explanation |
|---|---|
| Women perceived guidelines to be restrictive of their choices | High confidence: There is cogent, adequate and relevant data to support this finding however, there are minor concerns regarding the methodology of some of the studies which provide the data [18, 26, 31]. |
| Women who perceived their relationship with a healthcare provider positively felt empowered to make choices. | Low confidence: Minor methodological concerns and whilst there is cogent and relevant data, it is only sufficiently rich to support this finding in 1 study, with vague support coming from 1 other study [26, 31]. |
| Women's perceptions of risks influenced the choices that they made regarding their labour and birth. | High confidence: There are only very minor methodological concerns and the data are sufficiently rich in most examples to support this interpretative finding [18–20, 24, 30, 31]. |

In the first theme, women are informed of the increased risk of complications but deny the association or try to normalise their weight. Much of this data comes from studies in which the participating women had co-existing gestational diabetes or BMI>40kg/m$^2$. These women attribute obesity-related risks to other aetiology, such as smoking, glucose metabolism, hereditary factors or to supposedly falsified scientific studies in which they don't believe [21, 27, 33].

Keely (2011) describes how pregnant women with obesity consider risk to be just as likely to happen to them as to any other member of the pregnant population[21]. Some women with obesity justify the belief that their weight does not elevate their risk of complications by referring to the good state of their health, despite their weight[19, 21]. Other women evidence their beliefs by looking to women with uncomplicated pregnancies in their social networks (e.g. friends, family, members of online forums)[21].

Whilst this theme was well-supported, it was not exclusively the case for women who were informed of the risks of pregnancy; some of whom managed this knowledge pragmatically and used it as a motivating factor[35].

**Lack of preparation for risk.** It was quite clear from the data that the increased risk of obesity-related pregnancy complications was often avoided by both women and their healthcare professionals [19, 20, 29, 32, 36–38]. This was despite women expecting the risks to be raised when they were counselled in early pregnancy [36, 38]. Discussion was often deferred unless the risks occurred, by which time the women realised it was too late to change their behaviours to improve their outcomes [29]. Women found themselves needing to research potential risks externally, from peers or from the internet [29].

In avoiding the topic of obesity-related risks, healthcare professionals provided women with false reassurance regarding the potential course of their pregnancy [20, 36, 38]. Eventual realisation of the risks, either because of external reading or being presented with the reality of the risks when they arose, caused shock amongst the women [18, 19, 37]. It is important to note that this was not exclusively the case and that there were examples of women who felt they had been adequately counselled in advance and given time to adjust their behaviours to minimise the risk [35].

**Fearful acceptance and inevitability.** Many women were aware of the risk of pregnancy and accepted these, but knowledge of the risks caused them to worry or panic, taking away the expected pleasure of pregnancy [19, 27, 31–33, 38, 40]. In some cases, this panic was caused by insensitive counselling led by healthcare professionals[38].

Despite counselling on lifestyle choices which have the potential to minimise pregnancy complications, many women still felt that the risks were either inevitable, or their occurrence was out of their control. This finding was only evident amongst women with BMI>40kg/m$^2$ [21, 23]

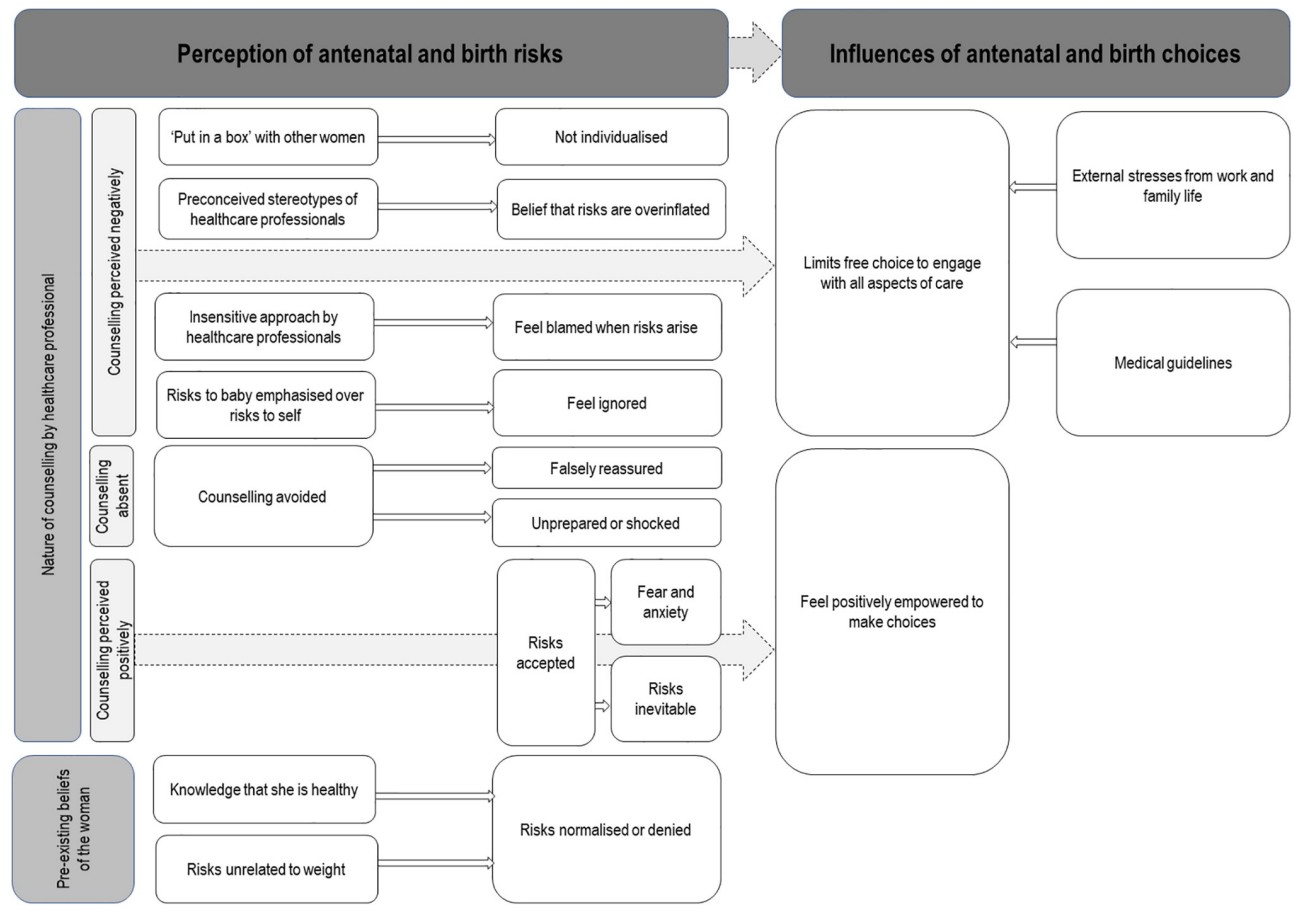

**Fig 2. Summary of relationship between qualitative findings.**

**Baby prioritised over the mother.** This final, smaller theme was discussed by multiple women in several studies. Women often felt forgotten about during the antenatal care, with the health and needs of their baby prioritised over their own [18, 30, 38].

### Study findings—Themes for influences of choices

With regards to our second research question–what influences the care choices that pregnant women with obesity make during pregnancy?—only two studies were identified which set out to elicit influencing factors for decision making during women with obesity's antenatal care [21] and intrapartum care [24]. Nevertheless, women interviewed in several other studies have commented on their influences when making choices during antenatal care and planning their birth wishes. Table 2 lists the types of choices which women felt they were presented with and raised during interviews for the included studies.

**External influences from personal stresses.** Factors within women's personal and professional lives affected the antenatal choices that they made. Women preferred the convenience of community appointments with flexible timing and closer location. Some women also felt restrained by social and economic stresses e.g. financial concerns, loss of employment or caring for children and older relatives. Similar external pressures also affected women's ability to make the recommended healthy lifestyle changes for pregnancy, or even to attend their appointments at all [18, 33].

**Table 2. Risks and choices discussed by women in the included studies.**

| ANTENATAL RISKS | LABOUR AND BIRTH RISKS | POSTNATAL RISKS |
|---|---|---|
| • *Gestational weight gain*<br>• *'Harm' to the baby*<br>• *'Big' baby*<br>• *'Blood clots' (venous thromboembolism)*<br>• *Difficulties visualising the fetus during scans*<br>• *Stillbirth*<br>• *Gestational diabetes*<br>• *High blood pressure* | • *'Difficult' labour*<br>• *Inability to move during labour*<br>• *Difficult/risky epidural insertion*<br>• *Shoulder dystocia* | • *Difficulties with breastfeeding*<br>• *Infant diabetes* |

| CHOICES FOR ANTENATAL CARE | CHOICES DURING BIRTH PLANNING |
|---|---|
| • Type of lead provider (midwife versus consultant).<br>• Whether to attend appointments<br>• Opportunity to change lead provider if the relationship was poor<br>• Place of antenatal care–midwifery-led community clinics versus hospital clinics.<br>• Timing of appointment (e.g. evening or weekend appointments for convenience).<br>• Whether to have glucose tolerance testing. | • Induction of labour<br>• Birthplace e.g. birth centre versus consultant-led service.<br>• Type of birth e.g. waterbirth, elective Caesarean section, 'natural" birth, vaginal birth after Caesarean (VBAC).<br>• Positions during labour<br>• Epidural analgesia<br>• Tubal ligation at Caesarean (sterilisation) |

**Restrictive guidelines.** Women perceived guidelines to be restrictive of both their antenatal and birth choices and frequently were the driving influences of decisions that they made [18, 26, 31]. Whilst they did not blame the healthcare professionals for this practice, they did remark that this led to loss of individual assessments and management plans [18, 26].

**Relationship with Healthcare Professional.** In most cases, a negative relationship with the HCP prevented women from feeling as if they had a choice in their antenatal care, including forcing them to disengage from routine care [26, 30, 31, 33]. Women were afraid to question decisions made by HCPs, who they felt stigmatised and over-medicalised them [26, 31, 33]. In another example, women felt coerced into making choices which suited the HCPs and were afraid of defying the healthcare professional [31].

Conversely, women who had a positive relationship with their HCP felt empowered by this and trusted in their ability to provide quality antenatal care [18, 26, 31]. Whilst there were examples of women who felt empowered by non-judgemental and supportive relationships with HCPs, there were no direct examples of how this led to them being supported in making choices around labour.

**Perception of risk.** Women's knowledge regarding current or past risks influences the choices that they made [18–20, 24, 30, 31]. Women who had previously experienced risk expected to have their choices limited in this subsequent pregnancy [31]. Where women were aware of risks, they were afraid of the outcomes and chose what they considered to be the safest route [20, 30]. Where risks were negatively presented, women were more likely to shy away from choices. In some cases, women felt confused about their choices because of contradictory counselling regarding risks [19, 30].

## Discussion

We have reviewed qualitative evidence documenting women with obesity's perceptions of risk during pregnancy and how this, and other factors, influence the choices they make for antenatal and intrapartum care. With high confidence, we have identified that women feel penalised

for their weight by being 'boxed in' with other women when counselled about risks and that an insensitive approach to risk-counselling can cause personal blame for potential complications. This association of weight stigma and personal blame is well-established in wider obesity literature [42, 43] and negative treatment and impersonal experience was also noted in a meta-synthesis on maternity experience by Smith et al (2011) [44], although this review did not set out to review how women perceive risks and make choices.

In previous qualitative work on risk perception in pregnancy, the paramount importance of the baby's safety is common [45–47]. This is also evident as a motivator for lifestyle modifications in pregnancies affected by maternal obesity [35, 40, 48, 49]. On the contrary, with high confidence, we have evidenced the phenomenon, that women with obesity perceive their baby's safety to overshadow their own psychological and physical wellbeing, leading them to feel neglected [18, 30, 38].

We conject with moderate levels of confidence the nature of counselling received by women with obesity influences their risk perception. Downe et al (2016) also found that the attitudes and behaviours of healthcare professionals were important to pregnant women [50]. Van Wagner (2016) documents the difficulty clinicians face in presenting risk to pregnant women in a balanced manner [51]. The choices that women with obesity make when planning for labour and birth are subsequently influenced by their perception of risk. Coxon et al (2017) similarly found that the choices of healthy women of normal weight were influenced by the views of healthcare professionals and the women's beliefs about risk[7].

We have also demonstrated that both women with obesity and healthcare professionals avoid discussion regarding complications in pregnancy. This falsely reassures the women and later, causes shock when complications occur. Smith (2011), Johnson (2013) and Jones (2017) have all concluded a similar finding through study of women with obesity or weight management in pregnancy [44, 52, 53] and Carter (2017) noted shock in women at risk of pre-eclampsia, who were not informed of the risk and required inpatient admission following routine antenatal appointments[54]. Unfortunately, the current literature is insufficiently rich to determine whether discussions are avoided because of the immateriality of risk, or the desire to avoid the stigmatising topic of obesity. The difficult balance between fully counselling women to increase awareness of early symptoms or behaviour modifications or causing anxiety regarding complications which may never arise is particularly important in the context of the duty of care to ensure women are fully informed regarding all potential consequences and choices. This is even more apparent in the UK following the Montgomery versus Lanarkshire Health Board court case[55], where Mrs Montgomery successfully won a civil Supreme court case against Lanarkshire Health Board for medical negligence because her obstetrician did not offer her an elective Caesarean section antenatally despite a big baby with a high risk of shoulder dystocia. The clinician did not offer this because they considered the risk of serious damage to the baby to be low and a Caesarean section to not be in the maternal interest. Unfortunately, the complication did arise, and the neonate was later diagnosed with cerebral palsy as a result.

With regards to influences of pregnancy and choices, we conject from the limited data that where women perceive their relationship with the healthcare provider negatively, they feel limited in the choices that they can make, either through fear to speak up or perceived coercion. Previous literature has identified that the way in which risk is framed, including risk where the potential is over-emphasised, may limit perceived choice[53, 56]. We evidenced that women perceive guidelines to be restrictive of their choices and whilst they understand this, they desire more individual assessments and an increase in offered choice. Coxon also argues that a health service facing increasing litigation has pushed clinicians towards strong adherence with guidelines, which affects women by a loss of autonomy[57]. What differs between our work and that

of Coxon, is that where women of normal weight can make choices and discuss the influences, women with obesity are unaware of their right to discuss personalised risks and benefits and to make choices. An evaluation of what influences the choices they make is therefore limited.

We set out to include all papers written in the English language, which does make the results less generalisable to countries where English is not primarily spoken—only three included studies arose from such settings, and to low or middle income countries—from which no studies were identified. Whilst two of the key findings were found to be relevant only to women with BMI$>40$kg/m$^2$, most of the studies included women with any class of obesity. We therefore consider the findings to be generalisable to all pregnant women with obesity in the UK, North America and Australasia, although this should be interpreted with caution in the context of women with obesity and co-existing social or medical risk factors. Since the included studies were conducted using women with obesity attending standard local antenatal care, these findings likely represent the feelings of women interacting with today's maternity care providers. This may not come as a surprise to some healthcare providers, yet there remains no evidence that style of counselling is changing for the better.

A common limitation of the included studies was that the primary researcher had not documented consideration of their personal bias when conducting the data collection and analysis. We have accounted for this in the GRADE-CERQual assessment of confidence but should also consider the bias of the primary researcher in this evidence synthesis, being an obstetrician with an academic interest in improving individualised care and greater choices for pregnant women with obesity. A thorough search of the included data has yielded very little evidence to refute the idea that women with obesity want more personalised care, and the assistance of a second researcher from a more general background to code the data and develop her own ideas has reduced this potential bias.

The limited data for synthesis was particularly problematic when assessing women with obesity's influences when making choices during pregnancy and birth. Traditionally in the UK (from where over half of the included studies arose), all women with obesity have been placed into high risk pathways, they therefore have had limited choices. In the light of recent guidance which permits clinicians and women with obesity more flexibility and choice regarding their lead carer and setting in pregnancy[4], further evidence is needed regarding the best methods to inform women with obesity regarding the potential risks of pregnancy in a sensitive and acceptable manner.

Further research should include how pregnant women with obesity respond to counselling and information provided in different formats (e.g. paper literature or mobile phone applications, group versus individual counselling), and by different health professionals. The potential for using decision tools or personalised risk calculators accounting for factors which confer better outcomes to pregnancy and birth is also of interest, enabling personalised and targeted counselling. Separate, robust research should be conducted to evaluate the safety of women with obesity but at otherwise low risk of complications being cared for in lower risk settings (e.g. multiparous women with uncomplicated pregnancy and birth being offered the choice of being cared in community clinics and giving birth in midwifery-led settings), and to identify the key influences when women with obesity make such choices antenatally or for birth planning.

## Supporting information

**S1 Table. ENTREQ checklist.**
(DOCX)

**S2 Table. Criteria used to define the components in the SPIDER literature search tool.**
(DOCX)

**S3 Table. Full search strategy and terms.**
(DOCX)

**S4 Table. Elements of the data extraction framework.**
(DOCX)

**S5 Table. Papers included and excluded following full text review.**
(DOCX)

**S6 Table. Characteristics of included studies.**
(DOCX)

**S7 Table. CASP Assessment of included studies.**
(DOCX)

**S8 Table. Full details of the GRADE-CERQual assessments.**
(DOCX)

## Author Contributions

**Conceptualization:** Sophie Relph, Dharmintra Pasupathy.

**Formal analysis:** Sophie Relph, Melissa Ong, Jane Sandall.

**Methodology:** Sophie Relph, Melissa Ong, Jane Sandall.

**Project administration:** Sophie Relph.

**Supervision:** Matias C. Vieira, Dharmintra Pasupathy, Jane Sandall.

**Writing – original draft:** Sophie Relph.

**Writing – review & editing:** Sophie Relph, Melissa Ong, Matias C. Vieira, Dharmintra Pasupathy, Jane Sandall.

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
