## [Decision Letter · Decision Letter 0]

10 Dec 2019

PONE-D-19-31821

Perceptions of risk and influences of choice in pregnant women with obesity: An Evidence Synthesis of Qualitative Research.

PLOS ONE

Dear Dr Relph,

Thank you for submitting your manuscript to PLOS ONE. After careful consideration, we feel that it has merit but does not fully meet PLOS ONE’s publication criteria as it currently stands. Therefore, we invite you to submit a revised version of the manuscript that addresses the points raised during the review process.

We would appreciate receiving your revised manuscript by Jan 24 2020 11:59PM. To enhance the reproducibility of your results, we recommend that if applicable you deposit your laboratory protocols in protocols.io, where a protocol can be assigned its own identifier (DOI) such that it can be cited independently in the future. For instructions see: http://journals.plos.org/plosone/s/submission-guidelines#loc-laboratory-protocols

We look forward to receiving your revised manuscript.

Kind regards,

Frank T. Spradley

Academic Editor

PLOS ONE

Journal Requirements:

1)

Reviewers' comments:

Reviewer's Responses to Questions

**Comments to the Author**

1. Is the manuscript technically sound, and do the data support the conclusions?

Reviewer #1: Yes

2. Has the statistical analysis been performed appropriately and rigorously? 

Reviewer #1: Yes

3. Have the authors made all data underlying the findings in their manuscript fully available?

Reviewer #1: Yes

4. Is the manuscript presented in an intelligible fashion and written in standard English?

Reviewer #1: Yes

5. Review Comments to the Author

Reviewer #1: The manuscript reports the findings of a review of qualitative research on perception of risk and childbirth choices for women with obesity using thematic synthesis. The subject matter is important and this report is timely and well conducted. The authors have used appropriate methods in the search, appraisal and synthesis of the included studies and the report is well-written.

I only have some small suggestions.

1. Use of the word 'influencers' feels a little awkward and does not add anything over 'influences' suggest change to influences.

2. Check the acronym HCP is defined at first use.

3. Line 97 sentence beginning 'In studies where women of weight in the non-obese range were included...' also feels a little awkward. I suggest the authors consider rewording.

Overall the paper was a pleasure to read and I look forward to seeing it in print.

6. PLOS authors have the option to publish the peer review history of their article (what does this mean?). If published, this will include your full peer review and any attached files.

Reviewer #1: Yes: Billie Bradford

---

## [Author Response · Author response to Decision Letter 0]

13 Dec 2019

Thank you for your comments and requests for minor edits. All changes have been made as suggested, please see the attached cover letter for response.

---

## [Editor Report · Decision Letter 1]

18 Dec 2019

Perceptions of risk and influences of choice in pregnant women with obesity. An Evidence Synthesis of Qualitative Research.

PONE-D-19-31821R1

Dear Dr. Relph,

We are pleased to inform you that your manuscript has been judged scientifically suitable for publication and will be formally accepted for publication once it complies with all outstanding technical requirements.

With kind regards,

Frank T. Spradley

Academic Editor

PLOS ONE

---

## [Editor Report · Acceptance letter]

23 Dec 2019

PONE-D-19-31821R1 

Perceptions of risk and influences of choice in pregnant women with obesity. An Evidence Synthesis of Qualitative Research. 

Dear Dr. Relph:

I am pleased to inform you that your manuscript has been deemed suitable for publication in PLOS ONE. Congratulations! Your manuscript is now with our production department. 

With kind regards,

on behalf of

Dr. Frank T. Spradley 

Academic Editor

PLOS ONE